# Thin-Layer Drying Model and Antifungal Properties of Rubber Sheets Produced with Wood Vinegar as a Substitute for Formic and Acetic Acids

**DOI:** 10.3390/polym17091201

**Published:** 2025-04-27

**Authors:** Wassachol Wattana, Putipong Lakachaiworakun, Natworapol Rachsiriwatcharabul, Visit Eakvanich, Panya Dangwilailux, Wachara Kalasee

**Affiliations:** 1Department of Engineering, King Mongkut’s Institute of Technology Ladkrabang, Chumphon Campus, Chumphon 86160, Thailand; wassachol.wa@kmitl.ac.th (W.W.); visit.ea@kmitl.ac.th (V.E.); panya.da@kmitl.ac.th (P.D.); 2Department of Sustainable Industrial Management Engineering, Faculty of Engineering, Rajamangala University of Technology Phra Nakhon, Bangkok 10800, Thailand; putipong.l@rmutp.ac.th (P.L.); natworapol.r@rmutp.ac.th (N.R.)

**Keywords:** fourier-transform infrared spectroscopy (FTIR), wood vinegar, ribbed smoked sheets (RSS), antifungal properties, drying kinetics

## Abstract

Currently, workers in the ribbed smoked sheet (RSS) rubber production industry face increasing health risks, primarily due to their direct involvement in converting fresh latex into raw rubber sheets. This process involves the manual addition of appropriately diluted commercial formic acid and acetic acid to induce coagulation, resulting in a tofu-like consistency, which is subsequently processed into rubber sheets. Previous studies have indicated that the use of commercial formic and acetic acids poses significant health hazards to workers and contributes to environmental pollution. Therefore, this study explores the feasibility of replacing commercial formic and acetic acids with wood vinegar derived from para-rubber wood, bamboo, and eucalyptus in the RSS production process. Wood vinegar samples from the three biomass sources were analyzed for their organic compound compositions using gas chromatography and subsequently used as coagulants in the preparation of raw rubber sheets. The drying kinetics and antifungal properties of the resulting sheets were then evaluated. The results revealed that wood vinegar derived from para-rubber wood contained the highest concentration of acetic acid (41.34%), followed by bamboo (38.19%) and eucalyptus (31.25%). Rubber sheets coagulated with wood vinegar from para-rubber wood and bamboo exhibited drying kinetics comparable to those obtained using acetic acid, with the two-term exponential model providing the best fit. Conversely, rubber sheets coagulated with eucalyptus-derived wood vinegar, which had a relatively high concentration of phenolic derivatives (22.08%), followed drying behavior consistent with the Midilli et al. model, similar to sheets treated with formic acid. In terms of antifungal properties, five fungal genera—*Aspergillus*, *Penicillium*, *Fusarium*, *Trichoderma*, and *Paecilomyces*—were identified on the rubber sheets. Fungal growth was most pronounced in the control samples (untreated with wood vinegar), whereas samples treated with wood vinegar exhibited significantly reduced fungal colonization. These findings indicate that wood vinegar is effective in inhibiting fungal growth on the surface of rubber sheets and may serve as a safer and more environmentally friendly alternative to commercial acid coagulants.

## 1. Introduction

Natural rubber, also known as para-rubber, plays a crucial role in various industries, including medical, aerospace, automotive, and public health sectors, as well as in the production of protective equipment. It is a naturally derived raw material and serves as a key feedstock for food-related and chemical industries [1,2]. Para rubber (*Hevea brasiliensis*) is one of Thailand’s most significant economic crops, ranking as the country’s third-largest export commodity after automobiles and computers. In 2023, Thailand had 24.1 million rai (approximately 3.86 million hectares) of rubber plantations, with an export volume of 2,866,467 metric tons [3,4]. The increasing demand for natural rubber is driven by the expansion of rubber-based product manufacturing industries [4], contributing to the growth of Thailand’s rubber exports. Before export and distribution, natural rubber undergoes processing, beginning with the collection of fresh latex from rubber plantations. The latex is filtered to remove impurities, coagulated with coagulants, and subsequently rolled into raw rubber sheets. Due to their high moisture content, these sheets must be dried to a moisture level of 3–5% (dry basis) to prevent fungal growth during storage, resulting in air-dried rubber sheets.

The acids commonly used as coagulants in the production of raw rubber sheets from fresh latex include acetic acid, formic acid, sulfuric acid, and ammonia mixed with acetic acid [5,6,7,8]. Studies on rubber sheets produced using fresh latex mixed with ammonia and acetic acid suggest that the concentration of ammonia–acetic acid should not exceed 0.15% to ensure that the latex concentration remains greater than that of the coagulant mixture. Additionally, storage duration has been found to affect the physicochemical properties of raw rubber sheets but does not significantly impact their mechanical properties [5,8]. However, the use of these acids in raw rubber sheet processing poses long-term health risks to farmers. Prolonged exposure, particularly through direct contact and repeated use, may lead to various health issues due to the presence of hazardous chemical compounds in these acids.

Wood vinegar, which contains over 200 different compounds, is produced through the pyrolysis of biomass during charcoal production [6,9,10,11]. As the smoke from this process cools, it condenses into wood vinegar, forming a clear brown liquid with a characteristic smoky odor. Based on previous studies, fresh latex coagulated into a tofu-like consistency using wood vinegar derived from bamboo, eucalyptus, and para-rubber wood, prior to being pressed into sheets, exhibits antifungal properties comparable to those of smoked rubber sheets coagulated with formic and acetic acids. Furthermore, the weight of rubber sheets produced using wood vinegar as a coagulant does not significantly differ from those produced using formic or acetic acid [5,9].

Currently, the cost of para-rubber wood firewood used as fuel in rubber smoking has significantly increased due to the expansion of the para-rubber wood furniture industry [12,13]. Moreover, the combustion of para-rubber wood firewood poses risks to human health and the environment, as it generates smoke particles and polycyclic aromatic hydrocarbons (PAHs), which are associated with cardiovascular diseases and cancer [13,14,15,16]. As a result, biogas from wastewater treatment ponds in rubber processing plants has been proposed as an alternative fuel [17]. However, since this approach is novel, challenges arise in storing biogas in compressed tanks. Due to these limitations, the research team initially tested liquefied petroleum gas (LPG) as a substitute. If the study yields promising and suitable results, further testing with biogas will be conducted. Previous studies have demonstrated that wood vinegar can replace formic and acetic acids in the coagulation of fresh latex into a tofu-like solid before processing it into raw rubber sheets. The resulting raw and dried rubber sheets exhibit chemical, physical, and mechanical properties comparable to those produced using formic and acetic acids [6,8,18]. According to the standards established by the Rubber Research Institute of Thailand (RRIT), air-dried sheets (ADS) are classified into five grades, ranging from Grade 1 (highest quality) to Grade 5 (lowest quality) [3,4]. This classification is primarily based on chemical and physical properties, color, the presence of impurities, and fungal contamination. Among these, impurities, fungal growth, and discoloration have a substantial impact on the market value of ADS rubber, as grading is determined by the degree of these defects. Excessive drying temperature and prolonged drying time can result in browning of the rubber sheets, thereby reducing their commercial value [1,2,6]. Consequently, effective quality control of ADS rubber necessitates a comprehensive understanding of drying kinetics and fungal resistance. However, no research has yet investigated the drying behavior of rubber sheets treated with wood vinegar as a substitute for formic or acetic acid using a thin-layer drying model to predict the drying process. Therefore, this study aims to analyze the chemical composition of wood vinegar, predict experimental drying outcomes using a thin-layer drying model, and investigate fungal species that develop on rubber sheets. Additionally, it examines the antifungal properties of wood vinegar derived from bamboo, eucalyptus, and para-rubber wood. The findings of this research are expected to contribute to the advancement of the rubber industry.

## 2. Materials and Methods

### 2.1. Raw Materials

In this study, fresh latex was obtained from RRIM 600 rubber trees (*Hevea brasiliensis*), with a total solid content (TSC) of 31.7 ± 0.8% and a dry rubber content (DRC) of 28.2 ± 0.5%, respectively. The TSC of fresh latex was determined using the moisture evaporation method (D1076:1988) [19], as specified by the American Society for Testing and Materials (ASTM). The DRC was measured by coagulating fresh latex with a 5% (*v*/*v*) acetic acid solution. The latex samples were sourced from the Agriculture, Food, and Energy Center at King Mongkut’s Institute of Technology Ladkrabang, Chumphon Campus, Pathio District, Chumphon Province, Thailand.

### 2.2. Commercial Acids and Wood Vinegar

In this study, acetic acid (99.8% *v*/*v*) and formic acid (94.0% *v*/*v*) were obtained from commercial suppliers in Thailand. For the production of raw rubber sheets using these acids, each was diluted with water to a final concentration of 2.5–3.0% *v*/*v* prior to mixing with fresh latex in the subsequent processing step [1,2,6]. Wood vinegar was produced via pyrolysis of bamboo, eucalyptus, and para-rubber wood using a prototype reactor previously designed and developed by the research team [20]. The resulting wood vinegar exhibited a dark yellow coloration, a characteristic smoky odor, and a specific gravity ranging from 1.005 to 1.050. Its principal constituents included acetic acid, methanol, furfuraldehyde, phenolic compounds, and neutral substances such as formaldehyde and acetone [6,20]. Due to its high content of volatile acids (10–30%), the wood vinegar exhibited acidic properties, with a pH ranging from 2.0 to 4.0. These acids contribute to its mildly corrosive nature. Nonetheless, wood vinegar is generally regarded as safe for humans, animals, plants, and the environment [6,20]. In the present work, wood vinegar derived from bamboo, eucalyptus, and para-rubber wood was utilized as a coagulant in the production of raw rubber sheets, serving as an alternative to conventional coagulants such as acetic acid and formic acid. The wood vinegar effectively induced the coagulation of fresh latex into a tofu-like consistency, which was subsequently rolled into raw rubber sheets approximately 3 mm in thickness. These sheets were air-dried for 12 h prior to final thermal drying. The pH of the wood vinegar was determined using a pH meter (Mettler Toledo, Greifensee, Switzerland). To analyze the chemical composition of the raw wood vinegar samples used in this study, 60 mL of each type of wood vinegar (bamboo, eucalyptus, and para-rubber wood) was prepared. Each sample was extracted three times with 50 mL of ethyl acetate using a separatory funnel. The ethyl acetate layers were collected and subsequently concentrated under reduced pressure using a rotary evaporator until the volume was reduced to 3 mL. The concentrated extracts were then diluted tenfold prior to injection into a gas chromatograph (Hewlett-Packard 5890 Series II, Waldbronn, Germany) for analysis. All analyses were performed in triplicate for each type of wood vinegar.

### 2.3. Characterization by Fourier-Transform Infrared Spectroscopy (FTIR)

In this study, FTIR spectra of all NR films were obtained using a Perkin-Elmer Spectrum GX FTIR spectrometer (PerkinElmer Inc., Wellesley, MA, USA). The NR films were prepared by evaporation casting method of 2% (*w*/*v*) NR solution in toluene, followed by solvent evaporation at room temperature for one week. The residual solvent was further removed by drying the films in a vacuum oven at room temperature for an additional week. The NR films were then analyzed using 32 scans at a resolution of 4 cm^−1^.

### 2.4. Drying Chamber and Procedure

The experimental drying chamber was constructed using 3.0 mm thick metal sheets and had dimensions of 60 × 40 × 250 cm^3^. Hot gases from LPG combustion were introduced into the chamber through two distribution ducts (10 cm in diameter) installed at the chamber floor. The temperature control system comprised sensors, a solenoid valve, and a relay controller to regulate the opening and closing of the LPG supply. In this experiment, rubber sheet samples were produced using wood vinegar derived from bamboo, eucalyptus, and para-rubber wood as a coagulant, replacing formic and acetic acids. Additionally, rubber sheets produced with formic and acetic acids were also tested. Nine sheets from each treatment were suspended on bamboo rods and dried using hot gases from the LPG combustion system, as illustrated in Figure 1.

All rubber sheet samples initially had a moisture content of 35.0 ± 2.5% on a dry basis. A centrifugal fan (Nitco, model RB60-520, Hessdorf, Germany) was used to maintain a constant airflow speeds of 0.5 m/s and 1.0 m/s. The ambient temperature, along with the inlet and outlet air temperatures used during drying, was measured using Type K thermocouples. Experiments were conducted five times at drying temperatures of 40 °C, 50 °C, and 60 °C, which are considered optimal for rubber sheet drying, as previously validated and accepted by researchers [1,2,6,7,12,21,22,23,24]. The drying process continued until the final moisture content reached 3.0 ± 0.5%, in accordance with the standards established by the Rubber Research Institute of Thailand (Bangkok, Thailand) [1,5,7,21]. At each drying condition, rubber sheet samples were removed from the drying chamber every six hours to measure their moisture content using the method specified by the American Society of Agricultural and Biological Engineers (ASAE) (St. Joseph, MI, USA) [2,8,21,22,23,24].

### 2.5. Thin-Layer Drying Models and Data Analysis

Thin-layer drying models used to describe the drying process of materials primarily comprise semi-theoretical and empirical drying equations, which are simplified solutions of Fick’s diffusion equation, as presented in Equations (1)–(5) in Table 1. Each model demonstrates variations in the slope of the moisture ratio over specific time intervals. The moisture ratio of rubber sheets is calculated using Equation (6).
polymers-17-01201-t001_Table 1Table 1Mathematical models used in the drying process.Model

Logarithmic MR=a⋅exp(−k⋅t)+b(1)Midilli et al. [25]MR=a⋅exp(−k⋅tn)+b⋅t(2)PageMR=exp(−k⋅tn)(3)Two-term exponentialMR=a⋅exp(−k⋅t)+(1−a)⋅exp(−k⋅a⋅t)(4)Wang and Singh [26]MR=1+a⋅t+b⋅t2(5)Where MR is the dimensionless moisture ratio, t is the drying time, a, b, k, n are the constant values.
(6)$$\mathrm{MR}=\frac{M-M_{\mathrm{eq}}}{M_{\mathrm{int}}-M_{\mathrm{eq}}}$$
where MR, M, M_eq_, M_int_ are the dimensionless moisture ratio, the moisture content at any time, the equilibrium moisture content, and the initial moisture content in dry basis percentage, respectively.

### 2.6. Evaluation of Wood Vinegar for Inhibiting Fungal Growth on Rubber Sheets

The experiment involved cutting rubber sheet samples into 10 × 10 cm pieces. Wood vinegar derived from bamboo, eucalyptus, and para-rubber wood was diluted with distilled water to concentrations of 10%, 20%, and 30% (*v*/*v*). Each rubber sheet sample was immersed in 100 mL of the prepared wood vinegar solution for 10 min and then air-dried at room temperature. The dried rubber sheet samples were stacked in three layers and stored in a container at 85 ± 1% relative humidity for four weeks. Following the storage period, fungal growth on the rubber sheets treated with different wood vinegar concentrations was assessed by counting fungal colonies and comparing them with untreated control samples.

### 2.7. Fungal Growth Inhibition Analysis

Fungal growth was quantified by measuring the surface area covered by fungal colonies using the standard plate count method. Physicochemical properties were analyzed according to FDA (2001) guidelines [27]. Rubber sheet samples were cut into 2 × 2 cm pieces and stacked in four layers. A 0.1% peptone solution (100 mL) was added, and the samples were shaken for 30 min to obtain a sample suspension. A 0.1 mL aliquot of this suspension was pipetted onto potato dextrose agar (PDA) plates and evenly spread until fully absorbed (approximately 10 min). The experiment was performed in triplicate, with no more than three stacked plates per incubation set. The plates were incubated at 20–25 °C for five days. The number of fungal colonies was then counted and reported as a reduction in colony-forming units per cm^2^ (CFU/cm^2^). The percentage inhibition of fungal growth was subsequently calculated by Equation (7).(7)$$\mathrm{Percentage}\;\mathrm{inhibition}\;\mathrm{of}\;\mathrm{fungal}\;\mathrm{growth}=\frac{M-N}{M}\times 100$$
where M is the average number of fungal colonies grown on the control culture medium and N is the average number of fungal colonies grown on the culture medium of the treated samples.

### 2.8. Statistical Analysis

In this study, analysis of variance (ANOVA) was utilized for graphical data analysis to identify interactions between response variables and estimate statistical parameters and process variables. The assumptions of the main model were evaluated using *p*-values at a 95% confidence level.

## 3. Results and Discussion

### 3.1. Chemical Composition of Wood Vinegar

The experimental results showed that the pH values of wood vinegar derived from para-rubber wood, bamboo, and eucalyptus were 3.77, 3.64, and 3.52, respectively. Table 2 presents the analysis of the 10 primary chemical components identified, including hydroxyacetone (2-propanone, 1-hydroxy), acetic acid, butanoic acid, propanoic acid, cyclotene, benzenemathanol, phenol, syringol (2,6-dimethoxyphenol), guaiacol (2-methoxyphenol), and p-cresol. Among these, para-rubber wood-derived wood vinegar exhibited the highest acetic acid concentration, followed by bamboo-derived and eucalyptus-derived wood vinegar. High concentrations of acetic acid and phenolic compounds have been identified as key indicators of wood vinegar quality, as they enhance antifungal properties by effectively inhibiting pathogenic fungal growth [5,9]. The findings in Table 2 indicate that the chemical compositions of the three types of wood vinegar vary in both concentration and composition, likely due to differences in the wood sources used for production. The primary components of wood vinegar are carbonization products derived from lignin, a major constituent of lignocellulose [4,9], which consists of phenylpropane units. The study found that para-rubber wood-derived wood vinegar contained the highest levels of acetic acid and phenolic compounds among the three types [5,9]. Consequently, it facilitated the coagulation of fresh latex into a tofu-like solid more effectively and provided superior fungal inhibition on rubber sheets compared to bamboo- and eucalyptus-derived wood vinegar.

### 3.2. The FTIR Spectra of NR Films

Figure 2 presents the FTIR spectra of NR films prepared using five different coagulants: wood vinegar derived from bamboo, eucalyptus, and para-rubber wood as substitutes for formic and acetic acids in rubber sheet production, along with rubber sheets produced using formic and acetic acids. The structural changes in =CH, CH_3_, and CH_2_ of cis-1,4-polyisoprene are indicated by absorption bands at 834, 1382, and 1453 cm^−1^. Previous studies have identified cis-1,4-polyisoprene as the primary component of natural rubber latex from para-rubber trees [9], which is corroborated by the absorption bands observed in this study. The FTIR results indicate that all NR coagulants produced NR films with highly similar structures and configurations. Additionally, the weight of NR sheets produced using wood vinegar from the three tested wood sources was comparable to that of NR sheets coagulated with formic and acetic acids. The pre-dried and fully dried NR sheets had weight ranges of 585–615 g and 525–535 g, respectively.

### 3.3. The Experimental Results for Drying Kinetics of Rubber Sheets

The moisture ratio of rubber sheets dried at air temperatures of 40, 50, and 60 °C under air velocities of 0.5 and 1 m/s is presented in Figure 3. The results indicate that, at a constant air velocity, an increase in drying chamber temperature enhances moisture evaporation from the rubber sheets. This effect is attributed to the higher kinetic energy of water molecules at elevated temperatures, which weakens intermolecular forces and facilitates moisture removal. Furthermore, at a constant temperature, an increase in air velocity results in a higher evaporation rate compared to lower air velocities. This phenomenon is due to the increased airflow, which promotes moisture removal from the rubber sheet surface and accelerates the drying process.

### 3.4. The Results for Thin-Layer Drying Models

The mathematical models for the thin-layer drying of NR sheets produced are presented with the results of the regression analysis conducted on the experimental data. The highest coefficient of determination (R^2^) and the lowest mean relative deviation (MRD) indicate the optimal fit of the mathematical model for drying all five types of rubber sheets examined in this study. The findings show that for drying at air temperatures of 40, 50, and 60 °C with an airflow velocity of 0.5 m/s, the R^2^ and MRD values range from 0.842 to 0.987 and from 0.011 to 0.081, respectively (see Table 3 and Table 4). In contrast, for drying at the same temperatures but with an airflow velocity of 1.0 m/s, the R^2^ and MRD values range from 0.835 to 0.982 and from 0.012 to 0.087, respectively (see Table 5 and Table 6). Among the five models evaluated (Table 1), the model proposed by the two-term exponential was identified as the most suitable for describing the thin-layer drying behavior of rubber sheets produced using wood vinegar derived from para-rubber wood, bamboo, and acetic acid (see Figure 4). The two-term exponential model exhibited the highest R^2^ values (0.942–0.987) and the lowest MRD values (0.011–0.028), respectively. This is attributed to the relatively high acetic acid content in bamboo and para-rubber wood compared to eucalyptus-derived wood vinegar. Conversely, the Midilli et al. model [25] was found to be the most suitable for characterizing the drying behavior of rubber sheets produced using eucalyptus-derived wood vinegar and formic acid (see Figure 5), yielding the highest R^2^ values (0.938–0.979) and the lowest MRD values (0.016–0.029), respectively. When acetic acid is used as a coagulant for fresh latex, the comparison between the thin-layer drying model and the experimental results in this study differs from the findings of Eakvanich et al. [8]. Their study reported that the logarithmic model more accurately predicted thin-layer drying behavior than the two-term exponential model. This discrepancy may be attributed to differences in the initial moisture content of the rubber sheets. In this study, the initial moisture content was 35.0 ± 2.5%, whereas in Eakvanich et al. [8], it was 80.0 ± 2.0%.

### 3.5. Effectiveness of Wood Vinegar in Inhibiting Fungal Growth on Rubber Sheets

The results, expressed in CFU/cm^2^, represent the number of fungal colonies observed on the rubber sheet surface (see Table 7). Five fungal genera were identified on the rubber sheets: *Aspergillus*, *Penicillium*, *Fusarium*, *Trichoderma*, and *Paecilomyces* (see Figure 6). Fungal growth was most pronounced in the control samples (rubber sheets untreated with wood vinegar), whereas samples treated with wood vinegar exhibited significantly lower fungal colonization. These findings indicate that wood vinegar effectively inhibits fungal growth on rubber sheet surfaces [5]. In addition, the results indicate that fungal growth on rubber sheets began after four weeks of storage at a relative humidity of 85 ± 1%. At a concentration of 10%, wood vinegar derived from eucalyptus, para-rubber wood, and bamboo inhibited fungal growth on raw rubber sheets by 97%, 91%, and 89%, respectively. At higher concentrations of 20% and 30%, all wood vinegar types exhibited strong antifungal activity. This effectiveness is attributed to the acidic nature of phenolic compounds and acetic acid present on the rubber sheet surface, which suppresses fungal growth. Among the tested concentrations, 30% wood vinegar demonstrated the highest antifungal efficacy, significantly outperforming the 20% and 10% concentrations (*p* < 0.05). Although increasing wood vinegar concentration enhances fungal inhibition, this study found that even at 10%, wood vinegar effectively suppressed fungal growth. Therefore, the use of wood vinegar at a 10% concentration represents a cost-effective alternative to higher concentrations, making it a viable option for the rubber industry to reduce chemical treatment costs while maintaining effective fungal control. Figure 7 presents the fungal resistance of rubber sheets treated with wood vinegar derived from eucalyptus wood, wood vinegar derived from para-rubber wood, wood vinegar derived from bamboo wood, acetic acid, and formic acid.

Recommendations: Previous studies on the physical properties of dried rubber sheets produced using wood vinegar as a substitute for formic and acetic acids in the coagulation of fresh latex into a tofu-like consistency—prior to processing into raw rubber sheets—have indicated that the resulting products exhibit only minor differences in tensile strength, Mooney viscosity, and plasticity retention index (PRI) compared to those produced using conventional coagulants [6].

## 4. Conclusions

The research findings indicated that the pH values of wood vinegar derived from para-rubber wood, bamboo, and eucalyptus were 3.77, 3.64, and 3.52, respectively. Among these, para-rubber wood vinegar contained the highest concentration of acetic acid (41.34%), followed by bamboo (38.19%) and eucalyptus (31.25%). Accordingly, rubber sheets coagulated with wood vinegar from para-rubber wood and bamboo exhibited drying kinetics comparable to those observed with acetic acid coagulation, best described by the two-term exponential model. In contrast, rubber sheets coagulated with eucalyptus-derived wood vinegar, which contained a relatively high concentration of phenolic derivatives (22.08%), followed drying kinetics consistent with the Midilli et al. model, similar to those treated with formic acid. With respect to antifungal activity, five fungal genera—*Aspergillus*, *Penicillium*, *Fusarium*, *Trichoderma*, and *Paecilomyces*—were identified on the rubber sheets. At a concentration of 10%, wood vinegar derived from eucalyptus, para-rubber wood, and bamboo inhibited fungal growth on raw rubber sheets by 97%, 91%, and 89%, respectively. Moreover, all three types of wood vinegar demonstrated enhanced antifungal efficacy at concentrations exceeding 20% and 30%, primarily due to the presence of phenolic compounds and acetic acid on the rubber surfaces, which effectively suppressed fungal proliferation. Although a 30% concentration of wood vinegar yielded the highest antifungal activity, from a practical standpoint—particularly regarding cost-effectiveness and sustained fungal control—a 10% concentration was deemed adequate for industrial application in the RSS production process. Future research should focus on the extraction and separation of individual compounds and acids present in wood vinegar to identify and evaluate the specific constituents responsible for significantly inhibiting fungal growth on rubber sheets. Such findings could contribute to reducing the production costs of dried rubber sheets for farmers and decreasing the reliance on chemical agents that may pose risks to human health and the environment.

## Figures and Tables

**Figure 1 polymers-17-01201-f001:**
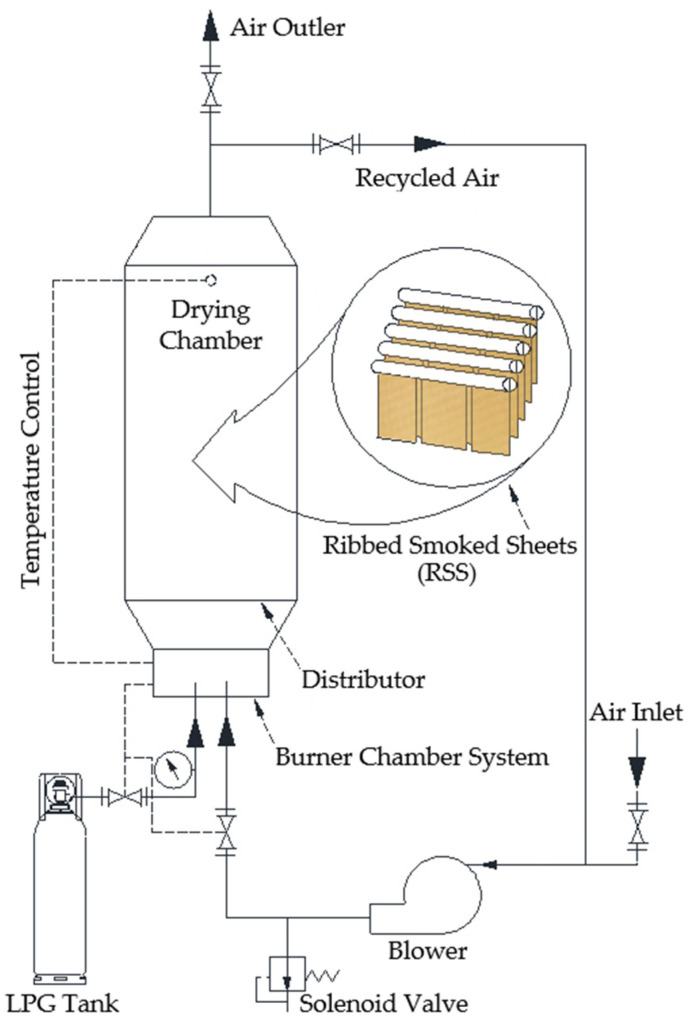
Schematic diagram of the drying system.

**Figure 2 polymers-17-01201-f002:**
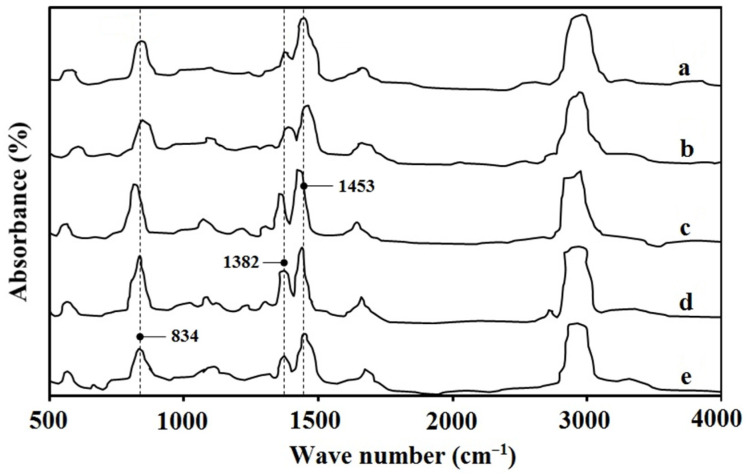
FTIR spectra of NR films coagulated by (a) acetic acid; (b) formic acid; (c) para-rubber wood vinegar; (d) eucalyptus wood vinegar; (e) bamboo wood vinegar.

**Figure 3 polymers-17-01201-f003:**
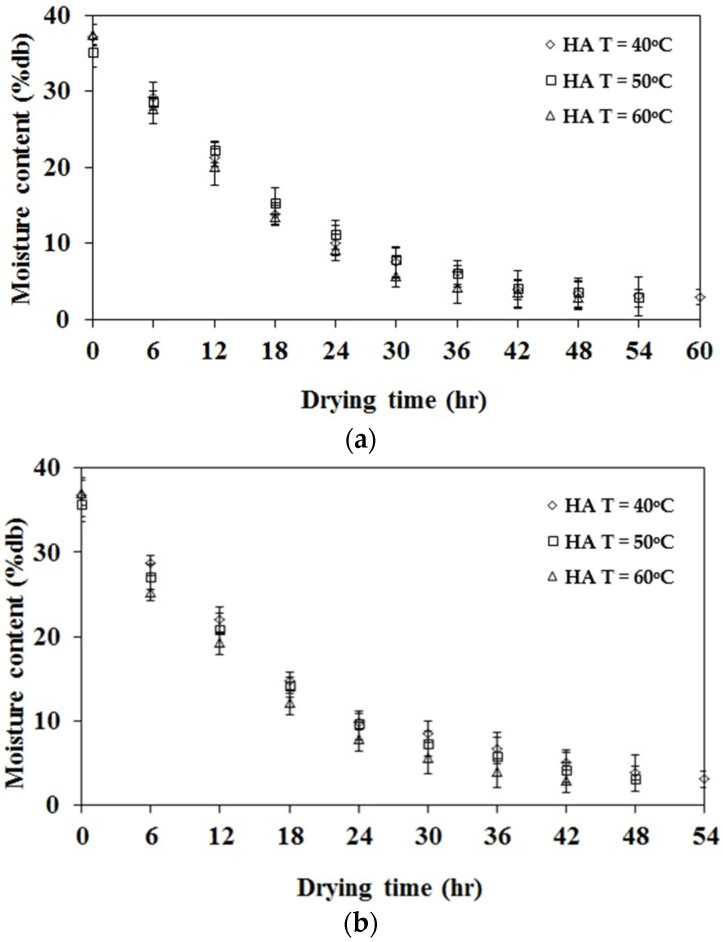
The moisture ratio of rubber sheets dried at air temperatures of 40, 50, and 60 °C. (**a**) Airflow velocities of 0.5 m/s. (**b**) Airflow velocities of 1 m/s.

**Figure 4 polymers-17-01201-f004:**
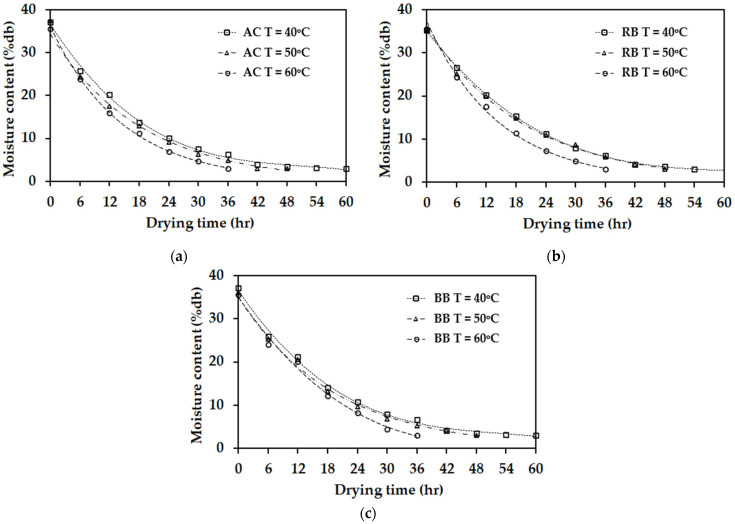
A comparative analysis of the two-term exponential models with the experimental results at each temperature. (**a**) acetic acid; AC. (**b**) para-rubber wood-derived vinegar; RB. (**c**) bamboo-derived vinegar; BB.

**Figure 5 polymers-17-01201-f005:**
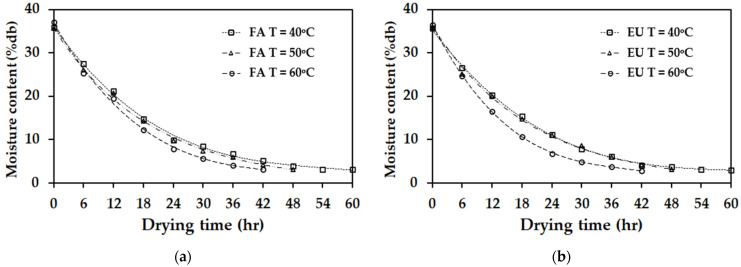
A comparative analysis of the Midilli et al. models with the experimental results at each temperature. (**a**) formic acid; FA. (**b**) eucalyptus-derived wood vinegar; EU.

**Figure 6 polymers-17-01201-f006:**
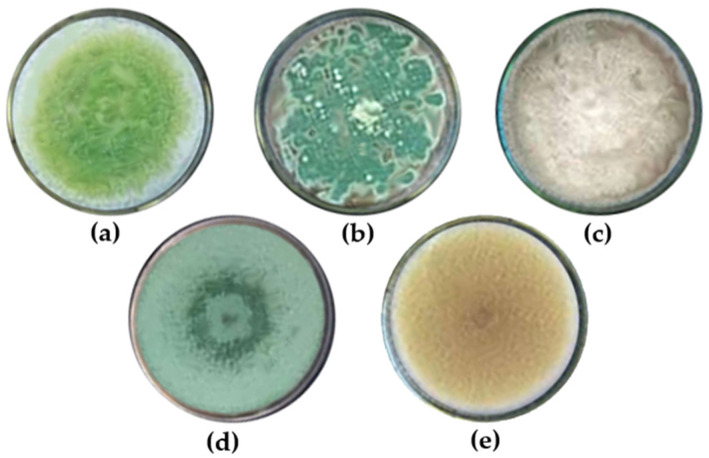
PDA analysis revealed the presence of five fungal genera on the rubber sheets: (**a**) *Aspergillus*; (**b**) *Penicillium*; (**c**) *Fusarium*; (**d**) *Trichoderma*; (**e**) *Paecilomyces*.

**Figure 7 polymers-17-01201-f007:**
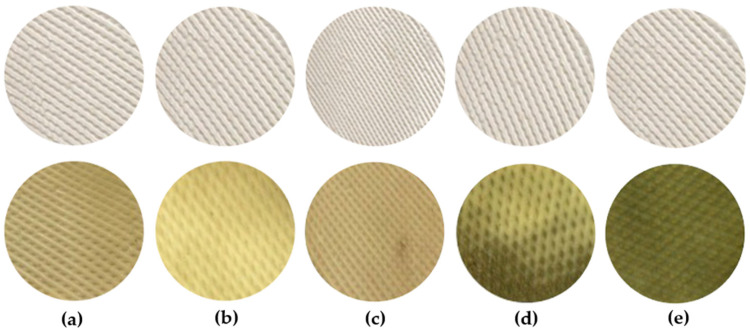
The fungal resistance of rubber sheets treated with wood vinegar derived and commercial acid (before and after drying process): (**a**) wood vinegar derived from para-rubber wood; (**b**) wood vinegar derived from bamboo wood; (**c**) wood vinegar derived from eucalyptus wood; (**d**) formic acid; (**e**) acetic acid.

**Table 2 polymers-17-01201-t002:** Volatile organic compounds in para-rubber, bamboo, and eucalyptus wood vinegar.

No.	Compound	Percentage of Total Area		
		Para-Rubber	Bamboo	Eucalyptus
1	Acetic acid	41.34	38.19	31.25
2	Phenol	8.29	7.56	7.12
3	2,6-dimethoxyphenol (Syringol)	7.38	5.53	11.07
4	2-Methoxyphenol (Guaiacol)	2.81	3.27	3.89
5	p-Cresol	1.57	1.72	1.85
6	Benzenemathanol	0.82	1.89	0.97
7	Cyclotene	2.79	2.15	2.38
8	Butanoic acid	0.95	1.18	0.83
9	Propanoic acid	2.13	3.14	1.75
10	2-propanone, 1-hydroxy (hydroxyacetone)	4.21	4.13	3.12
Total		72.29	68.76	64.23

**Table 3 polymers-17-01201-t003:** Parameters of the mathematical models for the thin-layer drying of NR sheets produced under air velocities of 0.5 m/s (wood vinegar derived).

Model	Parameters and Goodness of Fit	Para-Rubber Wood-Derived Vinegar			Bamboo Wood-Derived Vinegar			Eucalyptus Wood-Derived Vinegar		
		Drying Temperature			Drying Temperature			Drying Temperature		
		40 °C	50 °C	60 °C	40 °C	50 °C	60 °C	40 °C	50 °C	60 °C
Logarithmic	a	1.024	1.263	1.108	1.027	1.585	1.131	1.347	1.091	1.052
	b	−0.051	−0.032	−0.077	−0.089	−0.072	−0.084	−0.104	−0.079	−0.089
	k	0.024	0.031	0.038	0.028	0.045	0.052	0.036	0.058	0.071
	r^2^	0.865	0.882	0.892	0.871	0.878	0.909	0.842	0.911	0.893
	MRD	0.069	0.050	0.039	0.065	0.062	0.038	0.081	0.037	0.052
Midilli et al. [25]	a	0.056	0.075	0.062	0.047	0.083	0.041	0.036	0.058	0.069
	b	0.034	0.043	0.038	0.022	0.053	0.019	0.017	0.035	0.041
	k	0.297	0.315	0.283	0.228	0.352	0.213	0.205	0.281	0.304
	n	1.265	1.421	1.154	0.927	1.625	0.752	0.748	1.152	1.395
	r^2^	0.904	0.871	0.907	0.902	0.893	0.878	0.973	0.979	0.971
	MRD	0.039	0.065	0.038	0.039	0.052	0.062	0.018	0.016	0.019
Page	k	0.082	0.112	0.067	0.045	0.083	0.098	0.057	0.118	0.091
	n	0.543	0.718	0.412	0.382	0.543	0.583	0.399	0.729	0.582
	r^2^	0.887	0.853	0.865	0.945	0.907	0.853	0.902	0.918	0.905
	MRD	0.048	0.074	0.069	0.027	0.038	0.074	0.039	0.031	0.040
Two-term exponential	a	0.287	0.415	0.532	0.891	0.306	0.254	0.286	0.308	0.218
	k	4.021	5.327	6.189	9.852	4.115	3.926	4.019	4.117	3.728
	r^2^	0.967	0.975	0.982	0.968	0.987	0.971	0.902	0.893	0.885
	MRD	0.022	0.018	0.015	0.021	0.011	0.019	0.039	0.052	0.049
Wang and Singh [26]	a	0.072	0.056	0.028	0.039	0.085	0.095	0.062	0.074	0.853
	b	0.629	0.497	0.379	0.422	0.781	0.832	0.593	0.631	0.983
	r^2^	0.902	0.897	0.873	0.893	0.887	0.902	0.865	0.941	0.897
	MRD	0.039	0.051	0.065	0.052	0.048	0.039	0.069	0.029	0.051

Where r^2^ is correlation coefficient and MRD is mean relative deviation, a, b, k, n are the constant values.

**Table 4 polymers-17-01201-t004:** Parameters of the mathematical models for the thin-layer drying of NR sheets produced under air velocities of 0.5 m/s (commercial acid).

Model	Parameters and Goodness of Fit	Commercial Formic Acid			Commercial Acetic Acid		
		Drying Temperature			Drying Temperature		
		40 °C	50 °C	60 °C	40 °C	50 °C	60 °C
Logarithmic	a	1.185	1.095	1.029	1.127	1.052	1.136
	b	−0.086	−0.088	−0.025	−0.069	−0.082	−0.065
	k	0.043	0.052	0.059	0.035	0.049	0.057
	r^2^	0.887	0.853	0.902	0.885	0.871	0.905
	MRD	0.048	0.074	0.039	0.049	0.065	0.040
Midilli et al. [25]	a	0.035	0.071	0.078	0.032	0.045	0.038
	b	0.015	0.041	0.047	0.013	0.021	0.018
	k	0.201	0.304	0.329	0.195	0.228	0.215
	n	0.742	1.395	1.512	0.739	0.925	0.758
	r^2^	0.967	0.938	0.956	0.893	0.886	0.842
	MRD	0.022	0.029	0.024	0.052	0.049	0.081
Page	k	0.105	0.091	0.068	0.034	0.056	0.108
	n	0.687	0.582	0.413	0.337	0.398	0.689
	r^2^	0.902	0.893	0.878	0.975	0.871	0.902
	MRD	0.039	0.052	0.062	0.031	0.065	0.039
Two-term exponential	a	0.262	0.251	0.492	0.177	0.218	0.395
	k	3.958	3.922	5.993	3.855	3.728	4.937
	r^2^	0.907	0.871	0.901	0.942	0.983	0.953
	MRD	0.038	0.065	0.038	0.028	0.014	0.026
Wang and Singh [26]	a	0.049	0.028	0.041	0.074	0.025	0.095
	b	0.461	0.379	0.425	0.631	0.367	0.832
	r^2^	0.842	0.842	0.909	0.882	0.892	0.935
	MRD	0.081	0.081	0.038	0.050	0.039	0.041

Where r^2^ is correlation coefficient and MRD is mean relative deviation, a, b, k, n are the constant values.

**Table 5 polymers-17-01201-t005:** Parameters of the mathematical models for the thin-layer drying of NR sheets produced under air velocities of 1.0 m/s (wood vinegar-derived).

Model	Parameters and Goodness of Fit	Para-Rubber Wood-Derived Vinegar			Bamboo Wood-Derived Vinegar			Eucalyptus Wood-Derived Vinegar		
		Drying Temperature			Drying Temperature			Drying Temperature		
		40 °C	50 °C	60 °C	40 °C	50 °C	60 °C	40 °C	50 °C	60 °C
Logarithmic	a	1.042	1.363	1.267	1.165	1.028	1.375	1.025	1.024	1.048
	b	−0.083	−0.031	−0.043	−0.081	−0.029	−0.039	−0.024	−0.051	−0.081
	k	0.067	0.067	0.065	0.049	0.053	0.068	0.058	0.044	0.065
	r^2^	0.903	0.885	0.897	0.842	0.918	0.905	0.975	0.865	0.853
	MRD	0.037	0.049	0.051	0.081	0.031	0.040	0.031	0.069	0.074
Midilli et al. [25]	a	0.041	0.038	0.081	0.038	0.072	0.089	0.043	0.059	0.065
	b	0.019	0.019	0.045	0.015	0.042	0.041	0.022	0.027	0.033
	k	0.213	0.211	0.332	0.197	0.305	0.338	0.217	0.281	0.327
	n	0.752	0.749	1.527	0.741	1.397	1.582	0.758	0.916	1.338
	r^2^	0.902	0.971	0.906	0.907	0.975	0.906	0.978	0.952	0.945
	MRD	0.039	0.019	0.039	0.038	0.031	0.039	0.016	0.026	0.027
Page	k	0.062	0.085	0.048	0.067	0.068	0.105	0.059	0.102	0.105
	n	0.385	0.489	0.312	0.388	0.413	0.687	0.395	0.681	0.688
	r^2^	0.902	0.842	0.873	0.893	0.905	0.889	0.885	0.897	0.878
	MRD	0.039	0.082	0.065	0.052	0.040	0.048	0.049	0.051	0.062
Two-term exponential	a	0.258	0.498	0.311	0.256	0.398	0.825	0.365	0.295	0.157
	k	3.925	6.009	4.128	3.912	5.007	9.657	4.369	4.035	2.956
	r^2^	0.952	0.942	0.972	0.965	0.975	0.948	0.897	0.918	0.902
	MRD	0.026	0.028	0.019	0.023	0.018	0.027	0.051	0.031	0.039
Wang and Singh [26]	a	0.031	0.065	0.077	0.052	0.087	0.092	0.075	0.085	0.078
	b	0.382	0.598	0.635	0.486	0.789	0.834	0.634	0.781	0.635
	r^2^	0.885	0.853	0.865	0.897	0.887	0.873	0.842	0.908	0.906
	MRD	0.049	0.074	0.069	0.051	0.048	0.065	0.081	0.038	0.039

Where r^2^ is correlation coefficient and MRD is mean relative deviation, a, b, k, n are the constant values.

**Table 6 polymers-17-01201-t006:** Parameters of the mathematical models for the thin-layer drying of NR sheets produced under air velocities of 1.0 m/s (commercial acid).

Model	Parameters and Goodness of Fit	Commercial Formic Acid			Commercial Acetic Acid		
		Drying Temperature			Drying Temperature		
		40 °C	50 °C	60 °C	40 °C	50 °C	60 °C
Logarithmic	a	1.185	1.099	1.372	1.135	1.029	1.091
	b	−0.086	−0.081	−0.038	−0.086	−0.025	−0.079
	k	0.043	0.049	0.071	0.053	0.059	0.058
	r^2^	0.887	0.862	0.842	0.908	0.902	0.911
	MRD	0.048	0.070	0.081	0.038	0.039	0.037
Midilli et al. [25]	a	0.037	0.051	0.063	0.082	0.096	0.078
	b	0.020	0.031	0.035	0.053	0.065	0.051
	k	0.215	0.329	0.324	0.351	0.422	0.343
	n	0.756	1.356	1.325	1.628	1.825	1.605
	r^2^	0.972	0.965	0.971	0.842	0.907	0.905
	MRD	0.019	0.023	0.019	0.081	0.038	0.040
Page	k	0.089	0.065	0.115	0.099	0.093	0.075
	n	0.452	0.412	0.726	0.585	0.592	0.421
	r^2^	0.887	0.842	0.934	0.853	0.893	0.907
	MRD	0.048	0.081	0.041	0.074	0.052	0.038
Two-term exponential	a	0.867	0.854	0.895	0.225	0.535	0.425
	k	9.726	9.712	9.863	3.738	6.187	5.371
	r^2^	0.909	0.897	0.935	0.983	0.971	0.987
	MRD	0.038	0.051	0.041	0.015	0.019	0.011
Wang and Singh [26]	a	0.089	0.105	0.097	0.041	0.028	0.057
	b	0.901	0.995	0.846	0.427	0.375	0.499
	r^2^	0.921	0.865	0.872	0.893	0.941	0.902
	MRD	0.025	0.069	0.067	0.052	0.029	0.039

Where r^2^ is correlation coefficient and MRD is mean relative deviation, a, b, k, n are the constant values.

**Table 7 polymers-17-01201-t007:** Colony-forming unit of fungi on rubber sheets after treatment with different concentrations of bamboo, para-rubber, or eucalyptus wood vinegar, storage for four weeks at room temperature.

No.	Samples	Fungal Count (CFU/cm^2^)
1	Control I: Acetic acids	4.53 ± 0.35 × 10^3 b^
2	Control II: Formic acids	4.55 ± 0.27 × 10^3 b^
3	10% *v*/*v* eucalyptus wood vinegar	135 ± 0.56 ^e^
4	10% *v*/*v* para-rubber wood vinegar	411 ± 1.24 ^c^
5	10% *v*/*v* bamboo wood vinegar	514 ± 1.72 ^a^
6	20% *v*/*v* eucalyptus wood vinegar	71 ± 1.82 ^c^
7	20% *v*/*v* para-rubber wood vinegar	82 ± 2.19 ^f^
8	20% *v*/*v* bamboo wood vinegar	93 ± 1.45 ^d^
9	30% *v*/*v* eucalyptus wood vinegar	25 ± 0.79 ^h^
10	30% *v*/*v* para-rubber wood vinegar	38 ± 1.85 ^g^
11	30% *v*/*v* bamboo wood vinegar	47 ± 1.67 ^d^

Different small letters within one column indicate a significant difference (*p* < 0.05).

## Data Availability

The original contributions presented in this study are included in the article. Further inquiries can be directed to the corresponding author.

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
