# Peer review of "Thin-Layer Drying Model and Antifungal Properties of Rubber Sheets Produced with Wood Vinegar as a Substitute for Formic and Acetic Acids"

_polymers, 2025, doi:10.3390/polym17091201_

Round 1
Reviewer 1 Report
Comments and Suggestions for Authors
- Authors introduced the need for refining RSS rubber production techniques for health and environmental safety. However, the originality and novelty of the paper should be more clearly articulated in the Abstract and Introduction. What is the state-of-the-art development in this industry and what is the difference between this paper’s main point and the already-existing refining techniques?
- Material parameters such as purity of acetic acid and formic acid should be explicitly labeled.
- Figure 2, it would be better to add guiding arrows for each signal peak.
- Please provide figures of prepared samples and contrast of samples before/after fungi challenge.
- Please provide the GC data for wood vinegar samples.
- In addition to drying kinetics and anti-fungi ability comparison, can authors please provide comparison of appearance and other physical characteristics/parameters?
Author Response
Comments 1: Authors introduced the need for refining RSS rubber production techniques for health and environmental safety. However, the originality and novelty of the paper should be more clearly articulated in the Abstract and Introduction. What is the state-of-the-art development in this industry and what is the difference between this paper’s main point and the already-existing refining techniques?
Response 1: Thank you for pointing this out. We agree with this comment. Therefore, we have explained more details for the state-of-the-art development, and the difference between this paper’s main point and the already-existing refining techniques in the Abstract and Introduction – page number 1–3, Line 14–40 (Abstract), and Line 93–103 (Introduction).
Comments 2: Material parameters such as purity of acetic acid and formic acid should be explicitly labeled.
Response 2: Agree. We have, accordingly, done to emphasize this point – page number 3–4, Line 122–149 (2.2. Commercial Acids and Wood Vinegar).
Comments 3: Figure 2, it would be better to add guiding arrows for each signal peak.
Response 3: Agree. We have, accordingly, done to emphasize this point – page number 7, (Figure 2).
Comments 4: Please provide figures of prepared samples and contrast of samples before/after fungi challenge.
Response 4: Agree. We have, accordingly, done to emphasize this point – page number 5, Line 172–178 (2.4. Drying Chamber and Procedure), page number 10,11 Line 317–346 (3.5. Effectiveness of Wood Vinegar in Inhibiting Fungal Growth on Rubber Sheets), page number 13,14, (Figure 6,7).
Comments 5: Please provide the GC data for wood vinegar samples.
Response 5: We have, accordingly, done to emphasize this point – page number 7, (Table 2).
Comments 6: In addition to drying kinetics and anti-fungi ability comparison, can authors please provide comparison of appearance and other physical characteristics/parameters?
Response 6: We have, accordingly, done to emphasize this point – page number 11, Line 341–346 (3.5. Effectiveness of Wood Vinegar in Inhibiting Fungal Growth on Rubber Sheets).
Reviewer 2 Report
Comments and Suggestions for Authors
In this research article, authors attempted to use wood vinegar as a natural substitute for formic acid and acetic acid. This manuscript is very interesting to read and abstract is poorly written.
Abstract: Please rewrite the entire abstract with significant finding. One or two lines of background information is enough.
Keywords: In appropriate key works, please remove the first key word from the list
Line 47, write “Hevea brasiliensis” as, “Hevea brasiliensis” , Line 103- do the same;
Introduction: Introduction is adequate. Please include latest references, preferably 2021 onwards. Introduction must be hypothesis driven. Write your hypothesis at the end of the introduction, before the objective of this study.
Materials and methods
Section 2.2 is not clear, please revise
Add references in the experimental section
Results and discussion
Results are original and some amendments required
Section 3.5: Please delete the following three lines, it is a repetition of methodology
To assess the antifungal efficacy of wood vinegar derived from bamboo, eucalyptus, 276
and para-rubber wood, samples were diluted with distilled water to concentrations of 277
10%, 20%, and 30%. The effectiveness of each wood vinegar type in inhibiting fungal 278
growth on rubber sheets was evaluated after a four-week storage period.
Line 281: Five fungal genera were identified on the rubber sheets? How authors identified strains? By microscopy?, please describe colony morphology, especially colour in this section.
Line 282 and 282: Please use italics
Please combine section 3.6 with section 3.5
Result is very interesting but abstract is vague
Conclusion: Please revise conclusion to support the theme of this research work.
Line 341: Contradictory statements/terms are not required here
Line 343: Again, italics missed
Table 7:
Write 4.53×103 ± 0.35b as, 4.53± 0.35×103 b
Write 4.55×103 ± 0.27b as, 4.55± 0.27bx103 b
References
Related references included
Author Response
Comments 1: In this research article, authors attempted to use wood vinegar as a natural substitute for formic acid and acetic acid. This manuscript is very interesting to read and abstract is poorly written.
Abstract: Please rewrite the entire abstract with significant finding. One or two lines of background information is enough.
Response 1: Thank you for pointing this out. We agree with this comment. Therefore, we have rewritten the entire abstract with significant finding– page number 1, Line 14–40 (Abstract).
Comments 2: Keywords: In appropriate key works, please remove the first key word from the list
Line 47, write “Hevea brasiliensis” as, “Hevea brasiliensis”, Line 103- do the same;
Response 2: Agree. We have, accordingly, done to emphasize this point – page number 1, Line 41 (Keywords), page number 2, Line 48 (Keywords), and page number 3, Line 113 (2.1. Raw materials).
Comments 3: Introduction: Introduction is adequate. Please include latest references, preferably 2021 onwards. Introduction must be hypothesis driven. Write your hypothesis at the end of the introduction, before the objective of this study.
Response 3: Agree. We have, accordingly, done to emphasize this point – page number 2,3, Line 93–102 (Introduction).
Comments 4: Materials and methods
Section 2.2 is not clear, please revise
Add references in the experimental section
Response 4: Agree. We have, accordingly, done to emphasize this point – page number 3,4, Line 122–149 (2.2. Commercial Acids and Wood Vinegar).
Comments 5: Results and discussion
Results are original and some amendments required
Section 3.5: Please delete the following three lines, it is a repetition of methodology
To assess the antifungal efficacy of wood vinegar derived from bamboo, eucalyptus, 276
and para-rubber wood, samples were diluted with distilled water to concentrations of 277
10%, 20%, and 30%. The effectiveness of each wood vinegar type in inhibiting fungal 278
growth on rubber sheets was evaluated after a four-week storage period.
Line 281: Five fungal genera were identified on the rubber sheets? How authors identified strains? By microscopy? please describe colony morphology, especially colour in this section.
Response 5: Agree. We have, accordingly, done to emphasize this point – page number 10,11 Line 317–346 (3.5. Effectiveness of Wood Vinegar in Inhibiting Fungal Growth on Rubber Sheets), page number 13,14, (Figure 6,7).
Comments 6: Line 282 and 282: Please use italics
Please combine section 3.6 with section 3.5
Result is very interesting but abstract is vague
Response 6: Agree. We have, accordingly, done to emphasize this point – page number 1, Line 14–40 (Abstract).
Comments 7: Conclusion: Please revise conclusion to support the theme of this research work.
Line 341: Contradictory statements/terms are not required here
Line 343: Again, italics missed
Response 7: Agree. We have, accordingly, done to emphasize this point – page number 10,11 Line 317–346 (3.5. Effectiveness of Wood Vinegar in Inhibiting Fungal Growth on Rubber Sheets).
Comments 8: Table 7:
Write 4.53×103 ± 0.35b as, 4.53± 0.35×103 b
Write 4.55×103 ± 0.27b as, 4.55± 0.27bx103 b
Response 8: Agree. We have, accordingly, done to emphasize this point – page number 13 (Table 7).
Comments 9: References
Related references included
Response 9: We have, accordingly, done to emphasize this point – page number 15,16 (References).
Reviewer 3 Report
Comments and Suggestions for Authors
This study explores the potential of substituting commercial formic and acetic acids with wood vinegar derived from para-rubber wood, bamboo, and eucalyptus to mitigate health and environmental risks in RSS rubber production. An important topic concerning human health has been examined. The novelty and significance of this work should be clarified, especially in the abstract and conclusion sections.
Detailed remarks about the text are as follows:
The abstract provides information about the importance of the study and the scope of work. However, it lacks specific details regarding the results obtained. Important data from the study should be included in the abstract.
Comprehensive details about the composition of Commercial Acids and Wood Vinegar used in the study are essential. The sampling procedure requires further clarification.
L 164-165: Clarify how the components and their respective ratios were determined. There is no explanation provided for how the volatile organic compounds were analyzed. Although data is presented in Table 2, the methodology for these analyses is not described in the methods section.
The meaning of the data shown in Table 2 should be explicitly clarified. How will the differences attributed to Para-rubber, Bamboo, and Eucalyptus be determined? This should be clearly stated.
It should be clarified how the temperatures used in the drying phase were determined.
The Results and Discussion section contains many tables, but the results lack an in-depth discussion. Explain the trends observed in your results within this section. Provide more interpretation.
The Conclusions section should be more concise, including considerations about the importance of the results, the greatest novelties of the findings, and the areas for future research in this field.
Author Response
Comments 1: This study explores the potential of substituting commercial formic and acetic acids with wood vinegar derived from para-rubber wood, bamboo, and eucalyptus to mitigate health and environmental risks in RSS rubber production. An important topic concerning human health has been examined. The novelty and significance of this work should be clarified, especially in the abstract and conclusion sections.
Response 1: Thank you for pointing this out. We agree with this comment. Therefore, we have rewritten the entire abstract with significant finding– page number 1, Line 14–40 (Abstract), and page number 14, Line 390–414 (conclusion).
Comments 2: Detailed remarks about the text are as follows:
The abstract provides information about the importance of the study and the scope of work. However, it lacks specific details regarding the results obtained. Important data from the study should be included in the abstract.
Response 2: Agree. We have, accordingly, done to emphasize this point – page number 1, Line 14–40 (Abstract).
Comments 3: Comprehensive details about the composition of Commercial Acids and Wood Vinegar used in the study are essential. The sampling procedure requires further clarification.
Response 3: Agree. We have, accordingly, done to emphasize this point – page number 3–4, Line 122–149 (2.2. Commercial Acids and Wood Vinegar).
Comments 4: L 164-165: Clarify how the components and their respective ratios were determined. There is no explanation provided for how the volatile organic compounds were analyzed. Although data is presented in Table 2, the methodology for these analyses is not described in the methods section.
Response 4: We have, accordingly, done to emphasize this point – page number 6, Line 227–245 (3.1. Chemical Composition of Wood Vinegar).
Comments 5: The meaning of the data shown in Table 2 should be explicitly clarified. How will the differences attributed to Para-rubber, Bamboo, and Eucalyptus be determined? This should be clearly stated.
Response 5: We have, accordingly, done to emphasize this point – page number 6 Line 227–245 (3.1. Chemical Composition of Wood Vinegar).
Comments 6: It should be clarified how the temperatures used in the drying phase were determined.
Response 6: Agree. We have, accordingly, done to emphasize this point – page number 5, Line 172–178 (2.4. Drying Chamber and Procedure).
Comments 7: The Results and Discussion section contains many tables, but the results lack an in-depth discussion. Explain the trends observed in your results within this section. Provide more interpretation.
Response 7: Agree. We have, accordingly, done to emphasize this point – page number 10,11 Line 317–346 (3.5. Effectiveness of Wood Vinegar in Inhibiting Fungal Growth on Rubber Sheets), page number 13,14, (Figure 6,7).
Comments 8: The Conclusions section should be more concise, including considerations about the importance of the results, the greatest novelties of the findings, and the areas for future research in this field.
Response 8: Agree. We have, accordingly, done to emphasize this point – page number 14, Line 390–414 (conclusion).
Round 2
Reviewer 1 Report
Comments and Suggestions for Authors
N/A
Reviewer 3 Report
Comments and Suggestions for Authors
In my opinion, the comments indicated in the previous review have been taken into account to a sufficient extent.
The article can be accepted for publication.